# Impact of Different Anti-Hyperglycaemic Treatments on Bone Turnover Markers and Bone Mineral Density in Type 2 Diabetes Mellitus Patients: A Systematic Review and Meta-Analysis

**DOI:** 10.3390/ijms25147988

**Published:** 2024-07-22

**Authors:** Md Sadman Sakib Saadi, Rajib Das, Adhithya Mullath Ullas, Diane E. Powell, Emma Wilson, Ioanna Myrtziou, Chadi Rakieh, Ioannis Kanakis

**Affiliations:** 1Chester Medical School, Faculty of Health, Medicine and Society, University of Chester, Chester CH1 4BJ, UK; sadman.saadi@gmail.com (M.S.S.S.); rajibdas121@gmail.com (R.D.); adhithyamullath@gmail.com (A.M.U.); e.wilson@chester.ac.uk (E.W.); i.myrtzioukanaki@chester.ac.uk (I.M.); 2Metabolic Bone Service, Robert Jones & Agnes Hunt Orthopaedic Hospital NHS Foundation Trust, Oswestry SY10 7AG, UK; diane.powell2@nhs.net (D.E.P.); chadi.rakieh@nhs.net (C.R.); 3The Metabolic Bone Research Group (MBRG), Robert Jones & Agnes Hunt Orthopaedic Hospital, Oswestry SY10 7AG, UK; 4The Metabolic Bone Research Group (MBRG), Chester Medical School, Chester CH1 4BJ, UK; 5Department of Musculoskeletal & Ageing Science, Institute of Life Course & Medical Sciences (IL-CaMS), University of Liverpool, Liverpool L7 8TX, UK

**Keywords:** type 2 diabetes mellitus, anti-hyperglycaemic drugs, bone mineral density, bone turnover markers, diabetic bone disease, fracture risk

## Abstract

Diabetic bone disease (DBD) is a frequent complication in patients with type 2 diabetes mellitus (T2DM), characterised by altered bone mineral density (BMD) and bone turnover marker (BTMs) levels. The impact of different anti-diabetic medications on the skeleton remains unclear, and studies have reported conflicting results; thus, the need for a comprehensive systematic review is of paramount importance. A systematic search was conducted in PubMed and the Cochrane Library. The primary outcomes assessed were changes in BMD in relation to different anatomical sites and BTMs, including mainly P1NP and CTX as well as OPG, OCN, B-ALP and RANK-L. Risk of bias was evaluated using the JADAD score. The meta-analysis of 19 randomised controlled trials comprising 4914 patients showed that anti-diabetic medications overall increased BMD at the lumbar spine (SMD: 0.93, 95% CI [0.13, 1.73], *p* = 0.02), femoral neck (SMD: 1.10, 95% CI [0.47, 1.74], *p* = 0.0007) and in total hip (SMD: 0.33, 95% CI [−0.25, 0.92], *p* = 0.27) in comparison with placebo, but when compared with metformin, the overall effect favoured metformin over other treatments (SMD: −0.23, 95% CI [−0.39, −0.07], *p* = 0.004). GLP-1 receptor agonists and insulin analogues seem to improve BMD compared to placebo, while SGLT2 inhibitors and thiazolidinediones (TZDs) showed no significant effect, although studies’ number cannot lead to safe conclusions. For BTMs, TZDs significantly increased P1NP levels compared to placebo. However, no significant differences were observed for CTX, B-ALP, OCN, OPG, and RANK-L between anti-diabetic drugs and metformin or placebo. High heterogeneity and diverse follow-up durations among studies were evident, which obscures the validity of the results. This review highlights the variable effects of anti-diabetic drugs on DBD in T2DM patients, emphasising the need for long-term trials with robust designs to better understand these relationships and inform clinical decisions.

## 1. Introduction

Diabetes mellitus (DM) is a chronic condition characterised by persistent hyperglycaemia due to insulin resistance or insulin deficiency [1]. By definition, basic pathogenesis involves insulin deficiency in type 1 DM (T1DM) and insulin resistance in type 2 DM (T2DM), both leading to inevitable long-term complications [2]. The most frequent microvascular complications of DM include diabetic retinopathy, nephropathy, and neuropathy while macrovascular complications affect the coronary, cerebral and peripheral vessels leading to increased risk of heart disease, stroke or peripheral vascular disease [3], respectively. Although DM management encompasses a spectrum of these complications, the deleterious effects on the skeletal system are often overlooked or underestimated and thus significantly compromise an individual’s quality of life due to the development of diabetic bone disease (DBD) [4]. It is known that more than 35% of patients with DM suffer from bone loss leading to an increased risk of fragility fractures with exacerbated morbidity, mortality and healthcare costs [5,6]. 

Bone remodelling is a crucial process in maintaining the structural and functional integrity of the skeleton [7,8]. The bone remodelling cycle is principally regulated by osteoclastic and osteoblastic activities, which need to be in fine balance. While osteoclasts resorb the bone tissue, osteoblasts secrete bone matrix, mainly comprised of collagen type I, where calcium phosphate crystals are deposited during mineralisation [9,10]. In DM, bone quality depends not only on bone mass but also on microarchitecture. While T1DM is associated with reduced bone mineral density (BMD) [11], as assessed by dual-energy X-ray absorptiometry (DEXA), T2DM patients are usually presented with normal or even increased BMD [12,13] compared to healthy individuals, but this appears to be linked to compromised bone microarchitecture, mainly due to non-enzymatic glycation of collagen type 1, which alters bone mechanical properties [14,15]. Therefore, both types of DM lead to an increased risk of bone fragility [15]. Besides DEXA, bone turnover markers (BTM) reflect bone cell activity and can be used as reliable biomarkers to assess bone metabolism [16] and monitor osteoporosis management [17]. These are mainly by-products of bone destruction during osteoclastic proteolysis, like the C-terminal telopeptide of type I collagen (CTX) and the N-terminal telopeptide of type I collagen (NTX), which function as bone resorption markers [18,19,20]. On the other hand, procollagen-1 N-terminal peptide (P1NP) levels reflect collagen maturation while osteocalcin (OCN) and bone-specific alkaline phosphatase (B-ALP) are secreted during mineralisation and indicate osteoblastic bone formation activity [18,21,22,23]. In addition, other factors secreted by bone cells have been used as useful indicators of bone remodelling, such as the receptor activator of NFkB ligand (RANK-L) and osteoprotegerin (OPG), which regulate osteoblast–osteoclast communication [24]. When considering these two biomarkers, the OPG/RANK-L ratio is a more important indicator of bone mass and, subsequently, fracture risk than assessing the two molecules individually, especially since there are contradictory findings regarding their effects on bone metabolism [25,26].

Interestingly, various antidiabetic drugs show a significant impact on bone homeostasis, which is reflected in BMD and BTM [27]. Although both metformin and thiazolidinediones (TZDs) have been shown to decrease insulin resistance, metformin demonstrates a slightly positive osteogenic effect by stimulating proliferation and differentiation in osteoblastic cell lines [28]; however, TZDs, such as rosiglitazone, may increase bone loss and fracture risk via peroxisome proliferator-activated receptor gamma (PPARγ) agonist activity negatively affecting bone formation [29,30,31]. The effect of sulfonylureas (SNU) and insulin on bones is thought to be neutral; although, it has been reported that insulin treatment indirectly affects bone outcomes by increasing the frequency of falls related to hypoglycaemia in elderly T2DM diabetic patients [32]. As dipeptidyl peptidase-4 (DPP-4) enzymes are expressed in the membranes of osteocytes and are involved in bone turnover and collagen synthesis, DPP-4 inhibitors (DPP-4i) may cause bone health improvement without any negative impact on the skeleton [33,34]. Glucagon-like peptide-1 receptor agonists (GLP-1 RA) play a role in bone metabolism, potentially resulting in decreased bone resorption markers with a neutral effect on bone formation [34,35]. In contrast, sodium-glucose cotransporter-2 (SGLT-2) inhibitors, which control hyperglycaemia by inducing increased glycosuric effects, were shown to have a neutral or slightly positive effect on bone turnover and BMD in the long term [36].

Although different antidiabetic drugs seem to have a range of diverse effects on bone metabolism, the reported findings are conflicting and should be considered with caution, especially since the underlying pathophysiology still remains unclear for DBD. Several factors have been implicated in the development of DBD, encompassing hyperglycaemia, oxidative stress, and the accumulation of advanced glycation end products [15]. While some studies have attempted to summarise the effects of the aforementioned antidiabetic drugs [37], to our knowledge, there is no comprehensive systematic review with a meta-analysis comparing the impact of the major classes of treatments on BMD and BTMs. Therefore, we sought to fill this gap and provide an up-to-date analysis of the current literature, focusing on T2DM, which affects the vast majority of diabetic patients. We performed two sets of comparisons, placebo versus anti-diabetic drugs and metformin versus other diabetic drugs, as metformin is the first-line pharmacotherapy for T2DM patients [38]. We hypothesise that the effects of the different anti-diabetic medications on BMD and BTMs will be diverse with an overall positive outcome.

## 2. Materials and Methods

This systematic review with a meta-analysis was carried out in accordance with the recommendations provided by the Preferred Reporting Items for Systematic Reviews and Meta-Analyses (PRISMA) [39,40]. 

### 2.1. Search Strategy

A systematic database search was conducted independently by M.S.S.S., R.D. and A.M. in PubMed and Cochrane libraries up until March 2024. The full search strategy and the search terms used are described in Appendix A. 

### 2.2. Inclusion and Exclusion Criteria

Studies were included for analysis based on the following inclusion criteria: (i) randomized control trials (RCT) and non-RCT studies; (ii) participants with T2DM; (iii) undergoing treatment with anti-diabetic medication (Insulin, sulfonylurea, DPP-4 inhibitors, GLP-1 receptor agonists, SGLT-2 inhibitors or thiazolidinediones) with placebo or metformin as comparators; (iv) trials written and published in English; and (v) trials conducted on human subjects. The exclusion criteria were (i) participants with T1DM; (ii) participants with advanced cardiac and renal comorbidities; (iii) outcome not related to comparing bone biochemical marker changes or BMD.

### 2.3. Data Extraction and Risk of Bias

All the selected studies were thoroughly checked for the appropriate data extraction, by three authors (M.S.S.S., R.D. and A.M.U.) who retrieved the data independently. Any disagreements between authors were resolved by two independent reviewers (I.M. and I.K.). The following data were extracted from the selected articles: first author and publication year, study design, sample size (intervention and control), inclusion criteria, treatment received by the intervention group, follow-up period, and measured outcomes. To evaluate the potential risk of bias and the methodological quality of each individual study that was included in this review, the JADAD score was applied [41].

### 2.4. Statistical Analysis

Meta-analyses were performed on the standardised mean differences (SMD) of BMD and bone turnover markers for anti-diabetic drugs and placebo or metformin arms with standard deviations. In cases where studies reported interquartile ranges (IQR), the formula ‘standard deviation (SD) = width of IQR/1.35’ was used to calculate the missing SDs [42]. The random-effects model and inverse-variance method were applied for statistical significance assessment, due to high heterogeneity among studies. The statistical heterogeneity of outcome measurements between different studies was evaluated using the intersection of confidence intervals (95% CI) and was expressed as Cochran’s Q (χ^2^ test) and I^2^ values. Data classification for heterogeneity was employed as follows: low heterogeneity was based on I^2^ ranging from 30% to 49%, moderate heterogeneity from 50% to 74% and high heterogeneity from 75% and above [43]. Subgroup analyses were conducted based on specific anatomical sites—lumbar spine, femoral neck, and total hip for BMD—as well as on the different anti-diabetic treatments where applicable. To determine the overall effect of the anti-diabetic drugs on BMD, CTX and P1NP, a Z-test was performed, with statistical significance evaluated against a predetermined alpha level of 0.05. Furthermore, subgroup differences were investigated using a Chi-square test, assessing whether variations in effect sizes existed among different anatomical sites and anti-hyperglycaemic agents. The significance level for all tests was set at *p* < 0.05. All analyses were conducted using the Cochrane Review Manager Version 5.4 (RevMan 5.4.1) software.

## 3. Results

### 3.1. Identified and Included Studies

The PRISMA flow chart shows the schematic process of the database search and selection of articles for the review and meta-analysis (Figure 1). From the PubMed and Cochrane library search, 309 articles were found related to the keywords. Those articles were sent to the citation manager EndNote to remove duplications. Twenty-eight duplicates were removed, and the remaining 281 articles were screened through their titles and abstracts to check the eligibility according to the inclusion/exclusion criteria. In this phase, 149 articles were removed as they did not match the eligibility criteria. In the next step, the remaining 132 articles were checked for full-text retrieval, and 8 articles did not have full access. Next, 124 articles with full-text access were screened again, and 105 articles were removed due to the following reasons: irrelevant outcomes, different interventions in the treatment group, ineligible participants, and secondary analyses. Finally, 19 articles were selected for the systematic review and meta-analysis [36,44,45,46,47,48,49,50,51,52,53,54,55,56,57,58,59,60,61].

### 3.2. Study Characteristics

A total of 19 randomized control trials, published from 2007 to 2023, were reviewed in this study, involving 4914 participants. The follow-up duration of the included studies was between 12 and 102 weeks. Nine trials were placebo-controlled [36,44,45,46,47,48,49,50,51], whereas 10 trials compared active comparators among the different anti-diabetic agents [52,53,54,55,56,57,58,59,60,61]. Among the T2DM participants, several trials compared efficacy including only post-menopausal women [54,57,58,61]. With regard to the anti-diabetic agents, the studies covered GLP-1 agonists, including exenatide [56,59], dulaglutide [44] and liraglutide [47], SGLT-2 inhibitors, including ertugliflozin [45,48] and dapagliflozin [36,50], thiazolidinediones, including pioglitazone [46,49,51,58,59,60], balaglitazone [49], rosiglitazone [52,53,55,61], along with metformin [51,52,53,54,55,57,58,60,61] and sulfonylureas (Glimepiride) [45]. To evaluate the outcomes of different anti-diabetic drugs on the skeleton, BMD [36,44,46,48,50,53,56,57,58,59,61], P1NP [36,45,46,47,48,50,52,53,54], OPG [55,56,60], OCN [57,59] and B-ALP [53,54,56], as markers of bone formation, as well as CTX [36,45,46,47,48,50,52,53,54,56,59,61] and RANK-L [56] measurements, as indicators of bone resorption, were compared among the studies.

### 3.3. Risk of Bias Assessment

The overall quality of the included studies was considered high. Three RCTs had a JADAD score of 5/5 [36,46,50], ten studies had a JADAD score of 4/5 [44,45,47,48,49,51,52,53,54,61], whereas six studies had a score of 3/5 [55,56,57,58,59,60] (Table 1, detailed scores in Appendix A). 

### 3.4. Body Mineral Density (Anti-Diabetic Drug vs. Placebo)

The main analysis (k = 8; 1376 subjects with T2DM) showed an overall statistically significant difference in BMD between the anti-diabetic drug and the placebo group (SMD: 0.76, 95% CI [0.40, 1.13], I^2^ = 96%, *p* < 0.0001) (Figure 2). Subgroup analyses showed a statistically significant increase in BMD with anti-diabetic drug administration compared to the placebo group in the lumbar spine (SMD: 0.93, 95% CI [0.13, 1.73], I^2^ = 98%, *p* = 0.02) and femoral neck (SMD: 1.10, 95% CI [0.47, 1.74], I^2^ = 96%, *p* = 0.0007), while in total hip, the increase did not reach statistical significance (SMD: 0.33, 95% CI [−0.25, 0.92], I^2^ = 96%, *p* = 0.27). All analyses were characterised by high degrees of heterogeneity (*p* < 0.00001). However, the differences in effect sizes between different anatomical sites were not statistically significant (Chi^2^ = 3.30, *p* = 0.19, I^2^ = 39.5%). 

Subgroup analysis based on the class of medications revealed that SGLT2 inhibitors and TZDs did not show a statistically significant effect on lumbar spine BMD compared to placebo (*p* = 0.32 and *p* = 0.13, respectively), whereas GLP-1 RA and insulin analogues demonstrated a significant difference in BMD favouring anti-diabetic drugs (*p* < 0.00001 and *p* < 0.0001, respectively) (Appendix A). However, it should be noted that for GLP-1 RA and insulin analogues, only one study was included, respectively, and therefore, we did not perform further detailed analysis.

### 3.5. Body Mineral Density (Anti-Diabetic Drug vs. Metformin)

We then compared the effects of metformin versus other antidiabetic pharmacotherapies on BMD at different sites of the skeleton. The main analysis (k = 8; 802 subjects) showed an overall statistically significant effect on BMD favouring the metformin group over the anti-diabetic drugs group (SMD: −0.23, 95% CI [−0.39, −0.07], I^2^ = 43%, *p* = 0.004) (Figure 3). Subgroup analyses showed a weak statistically significant increase in BMD for the metformin group only in the femoral neck (SMD: −0.29, 95% CI [−0.58, 0.00], I^2^ = 45%, *p* = 0.05), while in the lumbar spine (SMD: −0.15, 95% CI [−0.48, 0.18], I^2^ = 59%, *p* = 0.39), as well as in the total hip (SMD: −0.28, 95% CI [−0.60, 0.05], I^2^ = 55%, *p* = 0.09), the differences were not significant. The analyses were presented with low (femoral neck, total) or moderate (lumbar spine, total hip) heterogeneity without any differences in effect sizes between different anatomical sites (Chi^2^ = 0.47, *p* = 0.79, I^2^ = 0%).

### 3.6. P1NP (Anti-Diabetic Drugs vs. Placebo)

The meta-regression analysis (k = 6; 1463 subjects) showed no significant difference in P1NP levels between individuals treated with anti-diabetic drugs and those receiving placebo (SMD: 0.09, 95% CI [−0.02, 0.20], I^2^ = 0%, *p* = 0.09) (Figure 4), and the heterogeneity was low (I^2^ = 0%, *p* = 0.56). In the subgroup analysis based on the class of drugs, thiazolidinedione showed a significant effect on P1NP levels compared to placebo (SMD: 0.44, 95% CI [0.01, 0.87], *p* = 0.04), but with only one study included, there were no significant differences observed for SGLT2 inhibitors or GLP-1 RA (Appendix A).

### 3.7. P1NP (Anti-Diabetic Drugs vs. Metformin)

In the analysis comparing metformin versus other antidiabetic drugs for P1NP (k = 4; 1923 participants), we found no significant difference (SMD: 0.04, 95% CI [−0.28, 0.36], I^2^ = 78%, *p* = 0.80) (Figure 5). However, high heterogeneity was detected, which highlights the presence of conflicting results in the literature.

### 3.8. CTX (Anti-Diabetic Drugs vs. Placebo)

With regard to CTX analysis, the overall effect size shows an SMD of 0.15 (95% CI: −0.02 to 0.31) favouring the anti-diabetic drugs (Figure 6), but the result is not statistically significant (Z = 1.75, *p* = 0.08). The heterogeneity test indicates a moderate variability among the studies (Chi^2^ = 9.58, *p* = 0.09; I^2^ = 48%). The subgroup analysis by class of medications also shows no statistically significant difference between the two groups for SGLT2 inhibitors, GLP-1 receptor agonists or thiazolidinedione, though the overall effect size favours the anti-diabetic drugs (Appendix A).

### 3.9. CTX (Anti-Diabetic Drugs vs. Metformin)

When comparing the effects of administrating metformin versus the other various antidiabetic drugs for CTX (k = 4; 1923 participants), we found no significant difference (SMD: 0.25, 95% CI [−0.02, 0.51], I^2^ = 67%, *p* = 0.07) (Figure 7). However, a trend in favouring the anti-diabetic drugs could be observed.

### 3.10. Other Bone Turnover Markers

Next, we aimed to analyse additional biomarkers of bone metabolism reported in the included studies; however, this was not feasible due to the inconsistency of the reported data with regard to comparators and the biomarkers measured (Table 2). One study reported a non-significant decrease in OCN by approximately 10% at 12 weeks in the group that received sitagliptin 100 mg/day [57], while Li et al. reported non-significant reductions in exenatide and pioglitazone groups and an increase in the insulin group [59]. Three studies compared the effect of anti-diabetic drugs and metformin on B-ALP, where one study reported a significant reduction in B-ALP levels with metformin compared to rosiglitazone [54], whereas two studies indicated no statistically significant difference in B-ALP levels between the two groups [53,56]. Esteghamati et al. compared metformin and pioglitazone for OPG levels and found a significant reduction only in men receiving pioglitazone [60]; similar findings were reported in Akyay et al. for the exenatide group as compared to insulin glargine with no changes for RANK-L [56]. 

## 4. Discussion

The present systematic review with a meta-analysis, including 19 studies, showed that different anti-diabetic medications have varying effects on BMD and bone turnover markers in individuals with T2DM. However, there is high heterogeneity among all studies, primarily attributed to different classes of anti-diabetic medications used in the studies, due to their different mechanism of action and their different direct effects on bone cells, which lead to diverse outcomes on BMD and BTMs [15].

The quantitative results and meta-analyses presented in this study provide valuable insights into the impact of anti-diabetic drugs on BMD. An overall statistically significant increase in BMD was observed for the group under anti-diabetic treatments in the lumbar spine, femoral neck as well as total hip as compared to placebo. A subgroup analysis based on a class of medications revealed that SGLT2 inhibitors and TZDs did not show a statistically significant effect on BMD, whereas GLP-1 receptor agonists and insulin analogues demonstrated a significant improvement in BMD. However, this result is inconclusive as there was only one study in the GLP-1 RA, insulin, and TZD group. This finding that SGLT2 inhibitors have no effect on bone density aligns with the previous meta-analyses and systematic reviews for canagliflozin, dapagliflozin, and empagliflozin, as well as real-world evidence in the literature, where no increased incidence of fractures [62,63,64] were reported. GLP-1 RAs do not appear to have an adverse effect on BMD. Their importance in bone metabolism appears to be linked to the differentiation of mesenchymal stromal/stem cells (MSC) into osteoblasts, which is promoted through the activation of GLP-1 receptors on the surface of bone marrow MSCs via PKA/β-catenin and MAPK pathways [65,66]. A systematic review including 38 studies reported a decreased incidence of bone fragility fractures using GLP-1 RAs [67], and another meta-analysis showed a reduced risk of hip fractures in the GLP-1 subgroup [62]. 

Even though there was no significant difference in P1NP and CTX levels in the anti-diabetic agents’ group, TZD showed a significant effect on P1NP levels compared to placebo. A meta-analysis by Zhu et al. indicated that the use of TZDs was associated with a decrease in BMD in the lumbar spine, total hip, and femoral neck, and an association, though not significant, was established between the cumulative use of TZDs and the risk of fractures [31]. In a meta-analysis of 138,690 people, insulin use was found to be positively associated with fracture risk [68]. This increased risk can be attributed to episodes of hypoglycaemia and the subsequent occurrence of falls [69]. In this review, only three studies included patients who received insulin showing a positive or neutral effect on bone metabolism [44,56,59]. Notably, other factors can interfere with the association between insulin use and fractures, such as the duration of the disease and the coexistence of complications, including diabetic microvascular disease, retinopathy, and neuropathy [70]. Our results also indicate that metformin appears to have either a slightly positive or neutral effect on bone biomarkers but is more impactful on BMD as compared to other anti-hyperglycaemic drugs. Similarly to our findings, a recent meta-analysis showed no statistically significant effect of metformin on BMD or BTMs [71]. Although BTMs are reliable and useful in monitoring bone turnover, it has been reported that they cannot be used as predictors for fracture risk in T2DM [72]. Overall, there is a lack of consistent results for BTM levels in DBD, highlighting the necessity of well-designed studies.

This review was subject to certain limitations, which should be noted. Due to the small number of available studies included and the diversity of the mechanisms of action of the interventions, it was not feasible to conduct more relevant subgroup analyses to identify potential effect modifiers and sources of heterogeneity, particularly in the analysis of GLP-1, insulin analogues, and TZDs. A high degree of heterogeneity was detected among the studies, implying that the findings of these analyses should be considered with caution. Most of the studies were conducted in high-income countries, which may limit the generalisation of our findings to other populations. In addition, the studies included in this systematic review have a diverse range of participant demographic characteristics as well as follow-up periods, which may influence the extrapolation of the results and the drawing of safe conclusions. Long-term trials that explore the impact of anti-diabetic agents on DBD should be designed in the near future. This would provide valuable insights into the sustainability of observed effects and the potential for adverse outcomes such as fractures.

## 5. Conclusions

In conclusion, this systematic review with a meta-analysis highlights the varied effects of different anti-diabetic medications on BMD and BTMs in T2DM individuals. The findings underline the complex interplay between anti-diabetic medications and bone metabolic homeostasis, with potential implications for fracture risk in individuals with T2DM. However, the small number of studies and the inconsistency in report results, as well as the high degree of heterogeneity, may affect the impact of the findings. Future research should focus on long-term trials with robust designs to elucidate the underlying mechanisms and long-term implications of these treatments on bone health, ultimately informing clinical decision-making and optimising patient outcomes. 

## Figures and Tables

**Figure 1 ijms-25-07988-f001:**
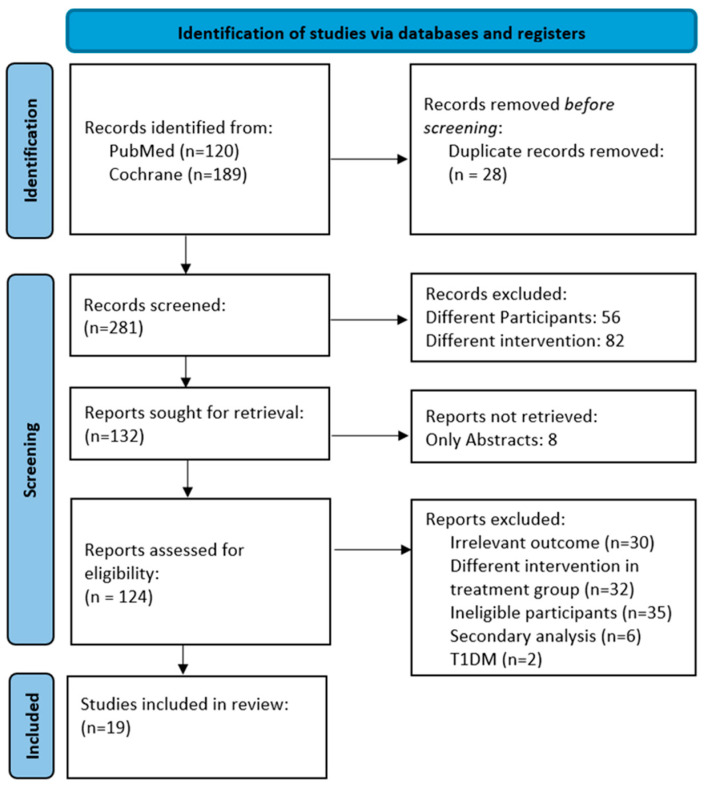
PRISMA flow diagram of the selection process for the included articles.

**Figure 2 ijms-25-07988-f002:**
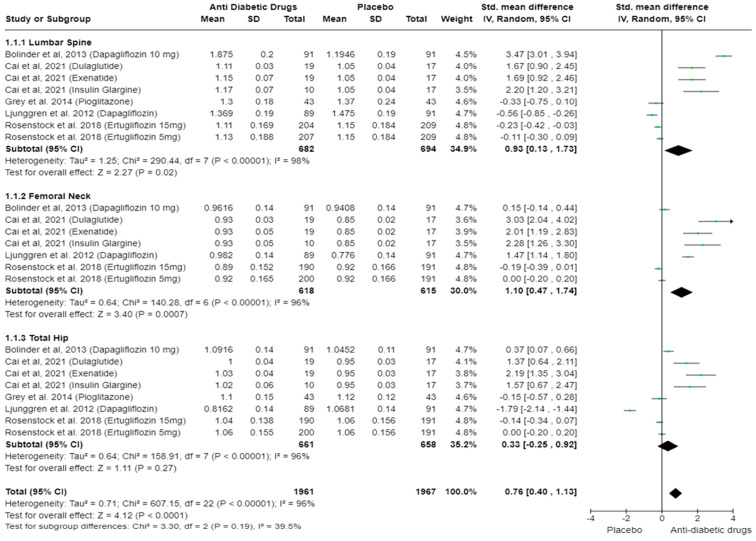
Standardised mean differences in BMD measurements based on different anatomical sites in T2DM patients receiving different classes of anti-diabetic drugs versus placebo. SMD in subgroups as well as total effect are presented with 95% confidence intervals using the random effects model [36,44,46,48,50].

**Figure 3 ijms-25-07988-f003:**
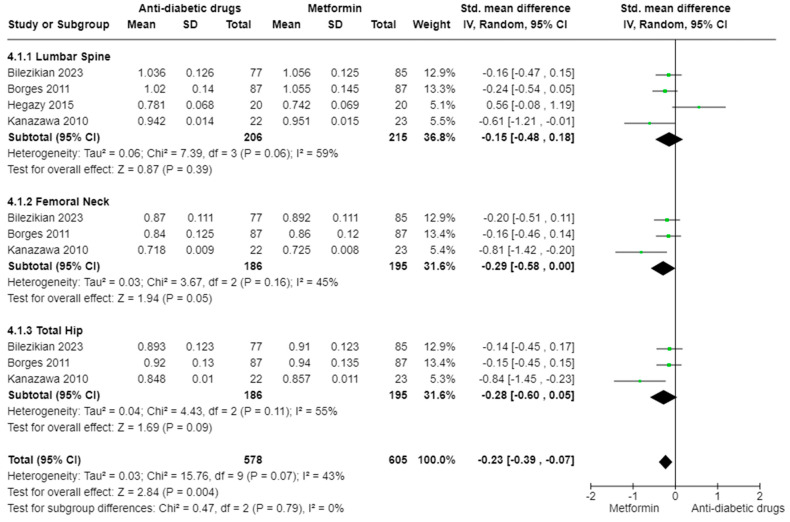
Mean differences in BMD measurements based on different anatomical sites in T2DM patients receiving different classes of anti-diabetic drugs versus metformin. Mean differences in subgroups as well as total effect are presented with 95% confidence intervals using the random effects model [53,57,58,61].

**Figure 4 ijms-25-07988-f004:**
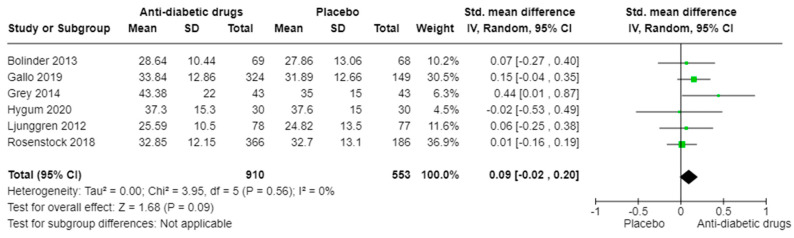
Mean differences in serum P1NP levels receiving various anti-diabetic drugs versus placebo. Mean differences in the studies as well as total effect are presented with 95% confidence intervals using the random effects model [36,45,46,47,48,50].

**Figure 5 ijms-25-07988-f005:**
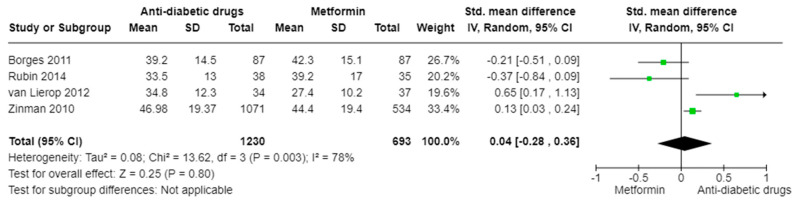
SMD in serum P1NP levels in patients receiving various anti-diabetic drugs versus metformin. Meta-regression was performed by applying the random effects model [51,52,53,54].

**Figure 6 ijms-25-07988-f006:**
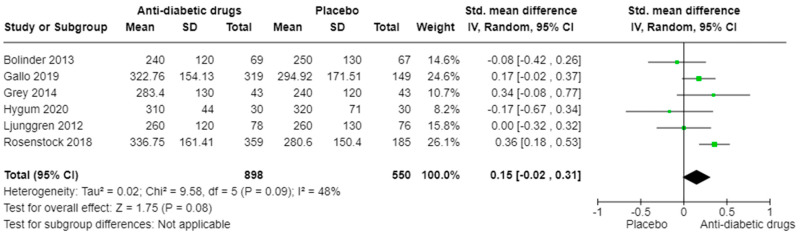
SMD in serum CTX levels comparing various anti-diabetic drugs versus placebo. Meta-regression was performed by applying the random effects model [36,45,46,47,48,50].

**Figure 7 ijms-25-07988-f007:**
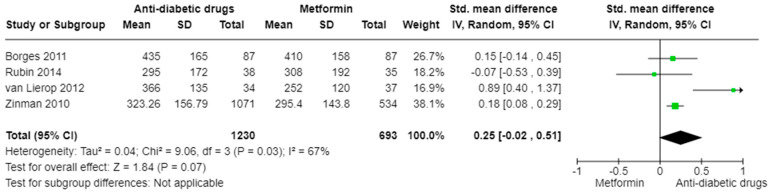
SMD in serum CTX levels in the comparison of anti-diabetic drugs versus metformin. Meta-regression was performed using the random effects model [51,52,53,54].

**Table 1 ijms-25-07988-t001:** Characteristics of the included studies.

First Author, Year	Country	Study Design	Sample Size	Mean Age	Dose Regimes	Follow-Up Period	Type of Diabetes	JADAD Score
			Intervention/Control	Intervention/Control				
Ljunggren et al., 2012 [36]	Multiple	Multicentre, parallel group, double-blinded, placebo-controlled RCT	Dapagliflozin (N = 89), Placebo (N = 91)	Dapagliflozin (60.6 ± 8.2) Placebo (60.8 ± 6.9)	Dapagliflozin 10 mg	50 weeks	T2DM	5/5
Cai et al., 2021 [44]	China	Single-blinded RCT	Exenatide (N = 19), Dulaglutide (N = 19), Insulin Glargine (N = 10), Placebo (N = 17)	Exenatide: (62.95 ± 1.70), Dulaglutide (57.42 ± 1.81), Glargine (64.36 ± 2.93), Placebo (62.00 ± 1.21)	Exenatide (2 mg/week), Dulaglutide (1.5 mg/week), Insulin glargine (6 unit/day), Placebo (once a week)	52 weeks	T2DM	4/5
Gallo et al., 2019 [45] (VERTIS MET extension)	Multiple	Double-blinded, placebo-controlled, 26-week treatment period (Phase A), and a double-blinded, 78-week treatment extension period (Phase B)	Ertugliflozin (N = 207), Ertugliflozin (N = 205), Placebo/Glimepiride (N = 209).	Ertugliflozin 5 mg (56.6 ± 8.2), Ertugliflozin 15 mg (56.9 ± 9.4), Placebo/Glimepiride (56.5 ± 8.7).	Ertugliflozin 5 mg, Ertugliflozin 15 mg, Placebo/Glimepiride.	104 weeks (26 weeks Phase A, 78 weeks Phase B)	T2DM	4/5
Grey et al., 2014 [46]	Australia, New Zealand	Double-blinded, placebo-controlled RCT	Pioglitazone (N = 43), Placebo (N = 43)	Pioglitazone (38–84) Placebo (48–79)	Pioglitazone 30 mg daily	12 months	T2DM	5/5
Hygum et al., 2020 [47]	Denmark	Double-blinded, placebo-controlled RCT	Liraglutide (N = 30), Placebo (N = 30)	Liraglutide (62 ± 8.38) Placebo (64 ± 3.83)	Liraglutide 1.8 mg daily	26 weeks	T2DM	4/5
Rosenstock et al., 2018 [48] (VERTIS MET trial)	Multiple	Multicentre, parallel group, double-blinded RCT	Ertugliflozin 5 mg (N = 207), Ertugliflozin 15 mg (N = 205), Placebo (N = 209)	Ertugliflozin 5 mg (56.6 ± 8.1), Ertugliflozin 15 mg (56.9 ± 9.4), Placebo (56.5 ± 8.7)	Ertugliflozin 5 mg daily, Ertugliflozin 15 mg daily	26 weeks	T2DM	4/5
Henriksen et al., 2011 [49]	Demark, Finland, Sweden	Phase 3, parallel group, multi-centred double-blinded RCT	Balaglitazone (N = 97), Balaglitazone (N = 97), Pioglitazone (N = 102), Placebo (N = 106)	Balaglitazone 10 mg (61.0 ± 8.8), Balaglitazone 20 mg (60.5 ± 9), Pioglitazone (60.1 ± 8.6) Placebo (60.9 ± 7.8)	Balaglitazone 10 mg daily, Balaglitazone 20 mg daily, Pioglitazone 45 mg daily	26 weeks	T2DM	4/5
Bolinder et al., 2013 [50]	Bulgaria, Czech Republic, Hungary, Poland and Sweden	Double-blinded, placebo-controlled RCT	Dapagliflozin (N = 91), Placebo (N = 91)	Women (55–75) Men (30–75)	Dapagliflozin 10 mg daily	102 weeks	T2DM	5/5
van Lierop et al., 2012 [51]	Netherlands	Prospective, double-blinded, two centred, parallel group RCT	Pioglitazone (N = 34), Metformin (N = 37), Placebo (N = 30)	Pioglitazone (56.5 ± 5.6) Metformin (55.0 ± 16.4)	Pioglitazone 30 mg daily, Metformin 1 gm twice daily	24 weeks	T2DM	4/5
Zinman et al., 2010 [52]	17 countries	Double-blinded, multi-centred, parallel group, RCT	Rosiglitazone (N = 549), Metformin (N = 551), Glyburide (N = 505)	Rosiglitazone (56.9 ± 10.0), Metformin (56.6 ± 9.4), Glyburide (66.7 ± 10.0)	Rosiglitazone 4 mg twice daily (N = 549), Metformin 1 gm twice daily (N = 551), Glyburide 7.5 mg twice daily (N = 505)	12 months	T2DM	4/5
Borges et al., 2011 [53]	9 countries	Phase 4. Multi-centred, double-blinded RCT	Rosiglitazone + Metformin (N = 344), Metformin (N = 334)	Rosiglitazone + Metformin (51.5 ± 10.5), Metformin (50.7 ± 10.5)	Rosiglitazone + Metformin 8 mg/200 mg, Metformin 2000 mg daily	80 weeks	T2DM	4/5
Rubin et al., 2014 [54]	Multiple	Multi-centred, double-blinded RCT	Rosiglitazone (N = 38), Metformin (N = 35)	Rosiglitazone (62.8 ± 6), Metformin (62.0 ± 5)	Rosiglitazone 8 mg daily, Metformin 2000 mg daily	52 weeks	T2DM (Post-menopausal women)	4/5
Nybo et al., 2011 [55]	Denmark	Multi-centred, double-blinded RCT	371 (Male 229, female 142)	56.2 ± 8.4	Rosiglitazone 8 mg daily, Metformin 2000 mg daily	24 months	T2DM	3/5
Akyay et al., 2023 [56]	Turkiye	Randomized, placebo-controlled, open-label, 2-arm parallel-group study	Exenatide (N = 15), Insulin Glargine (N = 15)	Exenatide (52.73 ± 4.68), Insulin Glargine (53.00 ± 4.07)	Exenatide 10 μg twice daily, Insulin Glargine 0.2 IU/Kg	24 weeks	T2DM	3/5
Hegazy et al., 2015 [57]	Egypt	RCT	Metformin (N = 20), Sitagliptin (N = 20)	58 to 66 years	Metformin 1000 mg daily, Sitagliptin 100 mg daily	12 weeks	T2DM (Post-menopausal women)	3/5
Kanazawa et al., 2010 [58]	Japan	Open-label RCT	Pioglitazone 15–30 mg/day (*n* = 22), Metformin 500–750 mg/day (n = 23)	Pioglitazone (67 ± 10), Metformin (66 ± 10)	Pioglitazone 15–30 mg/day, Metformin 500–750 mg/day	12 months	T2DM (Post-menopausal women)	3/5
Li et al., 2015 [59]	China	Two centres, parallel group RCT	Exenatide (N = 20), Pioglitazone (N = 21), Insulin (N = 21)	Exenatide (45.7 ± 10.5), Pioglitazone (51.3 ± 8.4), Insulin (53.0 ± 10.9)	Exenatide 10 μg twice daily, Pioglitazone 45 mg once daily, Insulin 0.4 IU/kg daily.	24 weeks	T2DM	3/5
Esteghamati et al., 2015 [60]	Iran	open-label, parallel-group, RCT	Metformin (N = 42), Pioglitazone (N = 46)	Metformin: (49.00 ± 1.66) (Women), (49.37 ± 2.06) (men). Pioglitazone (53.52 ± 1.57) (women), (49.54 ± 1.98) (men)	Metformin 1000 mg daily, Pioglitazone 30 mg daily	12 weeks	T2DM	3/5
Bilezikian et al., 2013 [61]	Multiple	Double-blinded, multicentred RCT	Rosiglitazone (N = 114), Metformin (N = 111)	Rosiglitazone (63.6 ± 6.61), Metformin (64.0 ± 6.46)	Rosiglitazone 8 mg daily, Metformin 2000 mg daily	52 weeks	T2DM (Post-menopausal women)	4/5

RCT: Randomised Control Trial; μg: microgram; IU: International Unit.

**Table 2 ijms-25-07988-t002:** Additional bone biochemical markers changes reported in the included studies.

First Author, Year	Intervention/Control with Dosage	Mean Changes from Baseline			
		OCN (Intervention/Control)	B-ALP (Intervention/Control)	OPG (Intervention/Control)	RANK-L (Intervention/Control)
Borges et al., 2011 [53]	Rosiglitazone + Metformin 8 mg/200 mg (N = 344) Metformin 2000 mg (N = 334)		Rosiglitazone + metformin: −27.1% Metformin: −20.9%		
Rubin et al., 2014 [54]	Rosglitazone 8 mg daily (N = 38) Metformin 2000 mg daily (N = 35)		Rosiglitazone: −9.2% Metformin: −22.9%		
Akyay et al., 2023 [56]	Exenatide 10 μg twice/day Insulin Glargine 0.2 IU/Kg		Exenatide: 3.4% Insulin Glargine: 7.6%	Exenatide: 45.07% Insulin Glargine: −9.1%	Exenatide: 40.1% Insulin Glargine: 31.4%
Hegazy et al., 2015 [57]	Metformin 1000 mg daily (N = 20) Sitagliptin 100 mg daily (N = 20)	Metformin −1.3 ± 0.9 μg/L Sitagliptin −2.05 ± 5.4 μg/L			
Esteghamati et al., 2015 [60]	Metformin 1000 mg daily (N = 42) Pioglitazone 30 mg daily (N = 46)			Metformin: Men −2.12 ± 1.62 pmol/L, Women −2.61 ± 1.08 pmol/L Pioglitazone: Men −1.18 ± 0.49 pmol/L, Women −0.39 ± 0.39 pmol/L	
Li et al., 2015 [59]	Exenatide 10 μg 2xdaily (N = 20), Pioglitazone 45 mg daily (N = 21), Insulin 0.4 IU/kg daily (N = 21)	Exenatide −0.619 ± 0.728 ng/mL Pioglitazone −0.150 ± 0.691 ng/mL Insulin 0.637 ± 0.787 ng/mL			
Henriksen et al., 2011 [49]	Balaglitazone 10 mg (N = 97), Balaglitazone 20 mg (N = 97), Pioglitazone 45 mg (N = 102), Placebo (N = 106)	Balaglitazone 10 mg 21% Balaglitazone 20 mg 18% Pioglitazone 14% Placebo 30%			
Nybo et al., 2011 [55]	Rosiglitazone 8 mg daily, Metformin 2000 mg daily			Change from baseline of 8 different randomized groups: NPH + placebo: 20 ± 343 ng/L, NPH + Metformin: 110 ± 319 ng/L, NPH + Rosiglitazone: −90 ± 531 ng/L, NPH + metformin+ Rosiglitazone: −100 ± 409 ng/L, Aspart+ placebo: 5 ± 274 ng/L, Aspart + Metformin: 80 ± 389 ng/L, Aspart + Rosiglitazone: −53 ± 427 ng/L, Aspart + Metformin + Rosiglitazone: −48 ± 318 ng/L	

## Data Availability

Not applicable.

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
