# Peer review of "Impact of Different Anti-Hyperglycaemic Treatments on Bone Turnover Markers and Bone Mineral Density in Type 2 Diabetes Mellitus Patients: A Systematic Review and Meta-Analysis"

_ijms, 2024, doi:10.3390/ijms25147988_

Round 1

Reviewer 1 Report

Comments and Suggestions for Authors

Comments to the Authors of manuscript number ijms-3110032 entitled “Impact of different anti-hyperglycaemic treatments on bone turnover markers and bone mineral density in type 2 diabetes mellitus patients: A Systematic review and meta-analysis

Diabetic bone disease (DBD) is a common complication in type 2 diabetes mellitus (T2DM), characterized by changes in bone mineral density (BMD) and bone turnover markers (BTMs). The impact of different anti-diabetic medications on bone health is unclear, with studies showing conflicting results. A systematic review and meta-analysis of 19 randomized controlled trials involving 4914 patients assessed changes in BMD and BTMs. Results showed that anti-diabetic medications generally increased BMD at various sites compared to placebo, but metformin was more effective overall. GLP-1 receptor agonists and insulin analogues improved BMD, while SGLT2 inhibitors, thiazolidinediones (TZDs), and metformin had no significant effect. TZDs increased P1NP levels, but no significant differences were found for other BTMs.

1. L 52-an increased risk of fragility fractures

2. L 65-an increased risk of bone fragility

3. L 94-95-what is unclear about the underlying pathophysiology of diabetic bone disease?

4. L 281-282-clarify why the different classes of anti-diabetic medications lead to high heterogeneity and how this impacts the study's conclusions?

5. This systematic review and meta-analysis provide valuable insights into the effects of various anti-diabetic medications on bone mineral density (BMD) and bone turnover markers (BTMs) in individuals with type 2 diabetes mellitus (T2DM). The comprehensive analysis highlights the differential impacts of anti-diabetic drugs, such as GLP-1 receptor agonists, insulin analogues, SGLT2 inhibitors, and thiazolidinediones, on bone health. Despite its strengths, the study also acknowledges significant heterogeneity among included studies and a limited number of available studies, particularly for specific drug classes.

6. The manuscript includes a broad range of anti-diabetic medications, offering a detailed comparative analysis of their effects on BMD and BTMs.

7. The findings are well-presented, with appropriate use of statistical methods to illustrate the results.

8. The discussion should emphasize the limitations related to the small number of studies for certain drug classes and their impact on the conclusiveness of the results.

Author Response

Please see attached document (Response to Reviewer 1)

Reviewer 2 Report

Comments and Suggestions for Authors

This manuscript involves a systematic review and meta-analysis of the effects of diabetic drugs on bone mineral density and bone turnover markers.

Abstract: Could you provide any numerical results in the abstract on your meta-analysis results (i.e., mean differences or standardized mean differences and confidence intervals). In one part of the abstract you imply that metformin was more effective than other drugs, and then you mention metformin was not effective. This seems contradictory.

I suggest you provide more detail on OPG and RANK-L in the introduction. Is an increase or decrease in these markers beneficial?

Could you provide some information in the introduction on the mechanisms by which bone is compromised in the diabetic condition?

Can you offer any hypothesis statements at the end of the introduction?

Please indicate whether the systematic review was registered with PROSPERO.

Section 2.2. In the inclusion criteria you need to indicate the comparators for included trials (i.e., placebo or metformin compared to other drugs).

In this section you indicate you excluded non-English studies, which can lead to bias. How many non-English studies were excluded? I suggest attempting to translate non-English studies.

Line 137: Please justify using a random-effects model instead of fixed-effects model in your meta-analyses.

Did you inspect funnel plots from your meta-analysis to determine publication bias?

Line 161: “8 articles had no full access” – did you attempt to retrieve these through inter-library loans or by contacting the authors?

Line 164: Please clarify what you mean by “secondary analyses” here.

Table 1: Provide the reference numbers (according to the reference section) for each study.

In table 1 you have provided the JADAD scores, but you could include more detail on this...i.e., which part of the JADAD scoring was high/low for each study (this probably necessitates a dedicated table).

Line 263: Please clarify why meta-analyses were not attempted for these comparisons. For example, line 269: “Three studies compared the effect of anti-diabetic drugs and metformin on B-ALP” – these seems sufficient for a meta-analysis.

Table 2: Again, I suggest including the citation numbers for the studies listed in this table, according to your reference list.

Author Response

Please see attached document (Response to Reviewer 2).
